# Simulations for the Locking and Alignment Strategy of the DRMI Configuration of the Advanced Virgo Plus Detector

Diego Bersanetti [1], Mattia Boldrini [2,3], Julia Casanueva Diaz [4], Andreas Freise [5,6], Riccardo Maggiore [5,6], Maddalena Mantovani [4] and Michele Valentini [7,8,*]

1. INFN, Sezione di Genova, I-16146 Genova, GE, Italy
2. INFN, Sezione di Roma, Piazzale Aldo Moro, 2-c/o Dipartimento di Fisica, Edificio, G. Marconi, 00185 Roma, RM, Italy
3. Department of Mechanical and Aerospace Engineering, Engineering Faculty, Università La Sapienza di Roma, Piazzale Aldo Moro, 5, 00185 Roma, RM, Italy
4. European Gravitational Observatory, Via E. Amaldi, 5, 56021 Cascina, PI, Italy
5. Department of Physics and Astronomy, VU Amsterdam, De Boelelaan 1081, 1081 HV Amsterdam, The Netherlands
6. Nikhef, Science Park 105, 1098 XG Amsterdam, The Netherlands
7. Department of Physics, University of Trento, Via Sommarive, 14, 38123 Trento, TN, Italy
8. Trento Institute for Fundamental Physics and Applications, Via Sommarive, 14, 38123 Trento, TN, Italy
* Correspondence: michele.c.valentini@gmail.com

**Abstract:** The Advanced Virgo Plus project aims to increase the sensitivity of the Virgo gravitational-wave detector, given the forthcoming O4 Observing Run. One of the major upgrades is the addition of the Signal Recycling Mirror in the optical layout. This additional mirror will provide a broadband improvement to the sensitivity curve of the instrument, but poses significant challenges in the acquisition and operation of the detector's working point. The process which brings the main optical components from the uncontrolled state to the final working point, which ensures the best detector sensitivity, is called lock acquisition: the lock acquisition is made by moving through increasingly more complex configurations toward the full control of all the interferometer's longitudinal degrees of freedom. This paper will focus on the control of the Dual-Recycled Michelson Interferometer (DRMI, the central part of the Virgo interferometer), presenting a comprehensive study of the optical simulations used in the design and the commissioning of this configuration. Treated topics include: the characterization of optical fields, powers, and error signals for the controls; the development of a trigger logic to be used for the lock acquisition; the study of the alignment sensing and control system. The interdependence between the three items has also been studied. Moreover, the validity of the studied techniques will be assessed by a comparison with experimental data.

**Keywords:** optics; controls; feedback; interferometry; simulations; gravitational-wave detectors; virgo; astrophysics

## 1. Introduction

The new Advanced Virgo Plus (AdV+) interferometer is currently being upgraded with the aim of reaching $\simeq$100 Mpc of Binary Neutron Star (BNS) range [1]. The upgraded interferometer will have a higher laser power, a frequency-dependent squeezed vacuum injected to reduce the quantum noise, and a more complex optical layout due to the addition of an Signal Recycling (SR) mirror.

The control strategy of the interferometer has been re-designed for the updated optical configuration. Indeed, in order to reach the extreme sensitivity ($\simeq 1 \cdot 10^{-18}$ m) required for the detection of gravitational waves, interferometric detectors require precise control of their main optical components. These optical components consist of mirrors and a beam splitter suspended by complex inertial isolation systems [2] that suppress seismic vibrations

leaving the mirrors as close as possible to a free-fall condition. Additionally, the position of the optics along the beam optical axis (also called *longitudinal* position) needs to precisely satisfy multiple cavity resonance conditions, defined in Section 1.1. This is achieved using continuous feedback control loops, reaching accuracy ranging from 10 nm to 1 pm RMS, depending on the involved optics.

Together with the overall control strategy, we implemented a new lock acquisition, which is the process of bringing the interferometer from an uncontrolled condition to its working point. This lock acquisition is based on the one already implemented in the Advanced Laser Interferometer Gravitational-Wave Observatory (LIGO) detectors [3]. The main difference with respect to the previous Advanced Virgo lock acquisition is that during most of the procedure, the Fabry-Pérot arm cavities are controlled by an auxiliary laser system that keeps them out of the resonance condition of the main laser [4]. The offset between the arm cavities resonance and the main laser frequency is commonly called "CARM offset" since it is applied to the Common ARM (CARM) length. When this CARM offset is applied, the impact of the Fabry-Pérot arm cavities on the main infrared laser (wavelength $\lambda = 1064$ nm) is minimal, and this allows to independently lock the central part of the interferometer, which consists of a Dual-Recycled Michelson Interferometer (DRMI) configuration, as shown in Figure 1. The Beam-Splitter (BS) and the input mirrors (North Input (NI), West Input (WI)) form a Michelson interferometer; the Power-Recycling (PR) mirror forms a resonant cavity with the input mirrors (the Power-Recycling Cavity (PRC)), and so does the SR mirror with the input mirrors (the Signal-Recycling Cavity (SRC)). The presence of two coupled optical cavities is challenging since the control of all the characteristic lengths has to be acquired *simultaneously*: this technique is called *coincidence lock*. This step is one of the main differences with respect to the previous "Variable-Finesse" lock acquisition [5], where instead the lock acquisition procedure was more entangled; and full characterization of it is therefore needed.

This document presents the simulation studies carried out to design and validate the lock of the DRMI. They can be divided into the following main topics:

- Choice and characterization of the longitudinal and angular error signals used to control the DRMI Degrees Of Freedom (DOFs);
- Determination of suitable trigger signals used for the engagement of the control loops with the "coincidence lock" technique;
- Study of the behavior of the longitudinal control loops' error signals when the DRMI mirrors are subject to misalignments.

### 1.1. The DRMI Configuration

To reach the desired working point, three relevant longitudinal DOFs can be defined for the lock of the DRMI: Power-Recycling Cavity Length (PRCL), Signal-Recycling Cavity Length (SRCL), and short MICHelson length (MICH). They are defined in Equation (1) according to the lengths depicted in Figure 1:

$$MICH = l_n - l_w \tag{1}$$

$$PRCL = l_P + \frac{l_n + l_w}{2} \tag{2}$$

$$SRCL = l_S + \frac{l_n + l_w}{2} \tag{3}$$

In addition to the longitudinal displacements, the laser beam inside the DRMI is affected by two rotational DOFs (for each mirror), called "TX" (or "pitch", i.e., around the mirror's horizontal axis) and "TY" (or "yaw", i.e., around the mirror's vertical axis).

The error signals that monitor the working point of the different DOFs are obtained using modulation and heterodyne demodulation techniques derived from the Pound-Drever-Hall (PDH) [6] and the Ward [7] techniques for the longitudinal and angular error signals, respectively. The laser beam is phase modulated at three different frequencies (6.27 MHz, 56.43 MHz and 8.36 MHz) using Electro-Optic Modulators (EOMs). The phase

modulation generates additional electromagnetic fields oscillating at the frequency of the main laser, plus or minus an integer multiple of the modulation frequency. These additional fields are also called *sidebands*. The modulation frequencies are chosen for the sidebands to satisfy specific resonance conditions inside the PRC and SRC (see next paragraph). In this way, their beating carries information about the different lengths in the interferometer, information that can be extracted from the several photodiodes installed at different ports of the interferometer: the signals are obtained both in DC and by demodulating at the same frequencies used for the initial modulation ("1f" signals), or their third multiple ("3f" signals). Section 2 specifies the choice of the many possible PDH signals for the control of the DRMI.

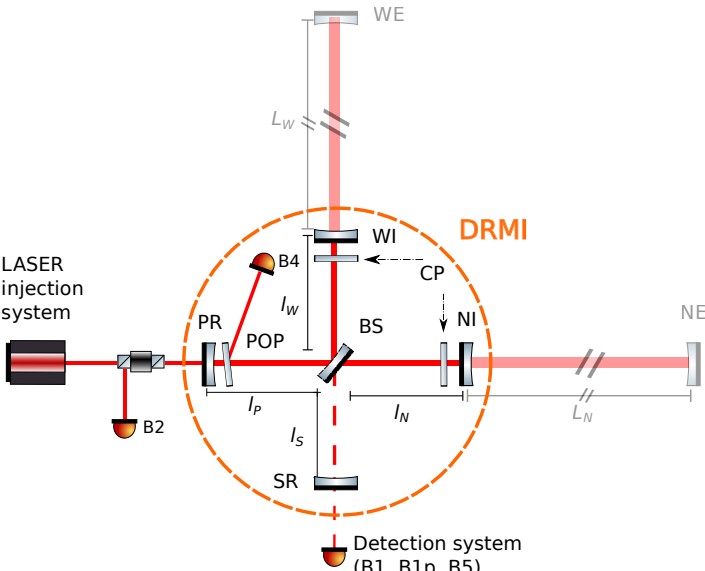

**Figure 1.** Simplified optical layout of AdV+, including the DRMI. The DRMI is circled in orange and includes the Power Recycling (PR), Signal Recycling (SR), North-Input (NI) and West-Input (WI) mirrors, and the Beam-Splitter (BS). The full interferometer, in its Dual-Recycled Fabry-Pérot Michelson Interferometer (DRFPMI) configuration, would also include the 3 km-long Fabry-Pérot arm cavities by the addition of the North-End (NE) and West-End (WE) mirrors. The figure also includes the position of the main photodetectors monitoring the DRMI beams: B2 (reflection from the PRC), B4 (intra-cavity pickoff of PRC), B1p (antisymmetric port pickoff), B1 (antisymmetric port beam after the output mode cleaner), B5 (stray-beam generated by the BS anti-reflective surface).

The working point of each longitudinal DRMI DOF while there is a CARM offset is defined by the following conditions:

- MICH: the short Michelson needs to be in dark fringe, i.e., with a complete destructive interference;
- PRCL: the Power Recycling cavity needs to be resonant with the 6 MHz and the 56 MHz sidebands, and anti-resonant with respect to the Carrier (once the CARM offset reduction is complete and the arm Fabry-Pérot cavities become resonant, the Carrier will become resonant in the PR cavity too due to the change in the phase of the cavities' reflected field);
- SRCL: the Signal Recycling cavity needs to be resonant with the 56 MHz sideband and resonant with respect to the Carrier (due to the same phase shift affecting the field in the PR cavity, at the end of the CARM offset reduction, the Carrier will become anti-resonant in the SR cavity).

The sideband resonance conditions in the recycling cavities are graphically represented in the left plot of Figure 2, which also shows how the 8 MHz one does not resonate in either cavity and is fully reflected by the interferometer.

After the CARM length offset is reduced to zero, the Carrier becomes resonant also in the Power Recycling cavity, as shown in the right plot of Figure 2.

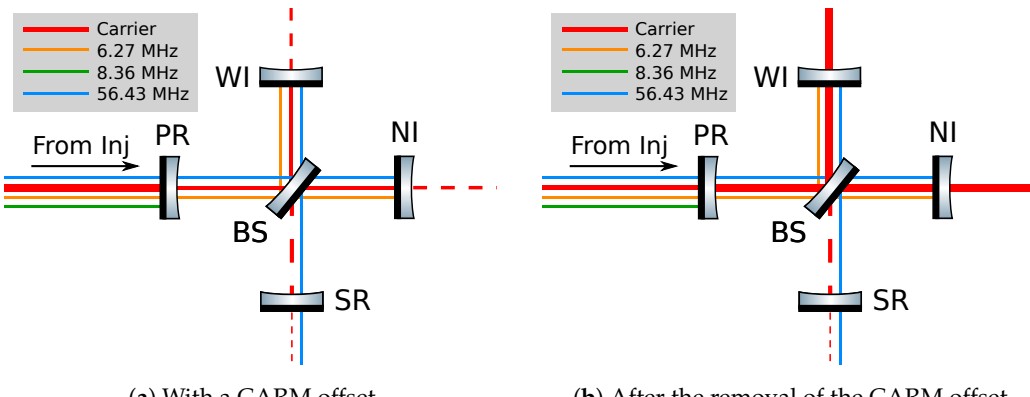

(**a**) With a CARM offset.

(**b**) After the removal of the CARM offset.

**Figure 2.** Resonance conditions of the control sidebands in the DRMI. The red line shows the path of the carrier beam, while the other colored lines correspond to the field of a specific sideband.

*1.2. General Setup of the Simulations*

Two simulation softwares have been used, FINESSE [8] (and its extension PYKAT [9]) and E2E [10], to carry out these studies from different points of view, frequency, and time domain, respectively. They require a model that represents as closely as possible the interferometer, including all the components that affect the electromagnetic laser field, but leaving out the components that do not directly affect the studied observables. For example, the models do not contain the detailed setup of each detection optical bench, the quantum noise reduction system, the pre-stabilized laser, or the injection systems since such inclusions would only add complexity. In addition to these simplifications, the interferometer has been considered in an optimally mode-matched condition, not considering possible mismatches. The main optical parameters used in the simulations are listed in Appendix A.

## 2. Longitudinal Error Signals of the DRMI

The first part of the study targets the choice of the best error signals for the control of the DRMI longitudinal DOFs. Since the control is done in coincidence, the target is to find signals with the highest possible Signal-to-Noise Ratios (SNRs), not to saturate the actuators. The aforementioned "1f" signals are the ones that provide the highest optical gain since they are based on the first harmonics of the sideband. The modulation frequencies have been each chosen to carry information about different DOFs: the 6 MHz sideband senses the PRC, while the 56 MHz senses both the PRC and the SRC. FINESSE simulations have been done by scanning the concerned DOFs around their working point, including all the available sensors (B2, B4, B1p, B5, shown in Figure 1). The signals coming from the B2 sensor (reflection of the PRC) were the ones having the highest SNRs. Regarding the Michelson length, simulations have shown that the quadrature of the B2 56 MHz signal is a suitable error signal, and its information is very well decoupled from the SR one. Table 1 summarizes the ideal sensing for the lock acquisition of the DRMI.

**Table 1.** Planned error signals for the longitudinal control of the DRMI DOFs.

| Lock Acquisition Step | PRCL | MICH | SRCL |
|---|---|---|---|
| Initial lock ("1f") | B2 6 MHz | B2 56 MHz I | B2 56 MHz Q |
| CARM offset reduction ("3f") | B2 18 MHz | B2 169 MHz I | B2 169 MHz Q |

These signals, however, are influenced by the carrier beam's phase shift, which happens when the Fabry-Pérot arm cavities are brought back to resonance as a consequence of the CARM offset reduction. After the initial lock, therefore, the control is handed off to a second set of signals, based on the "3f" demodulation of the original 1f-modulated signals. These signals only provide information carried by sidebands, so they are robust against the carrier phase shift. As seen in Figure 3, they have a lower gain and, consequently, a higher sensing noise, which is why they are not used for the lock acquisition but only after the initial lock.

Both the first and the second set of error signals have been identified by the FINESSE modal simulations reported in [11].

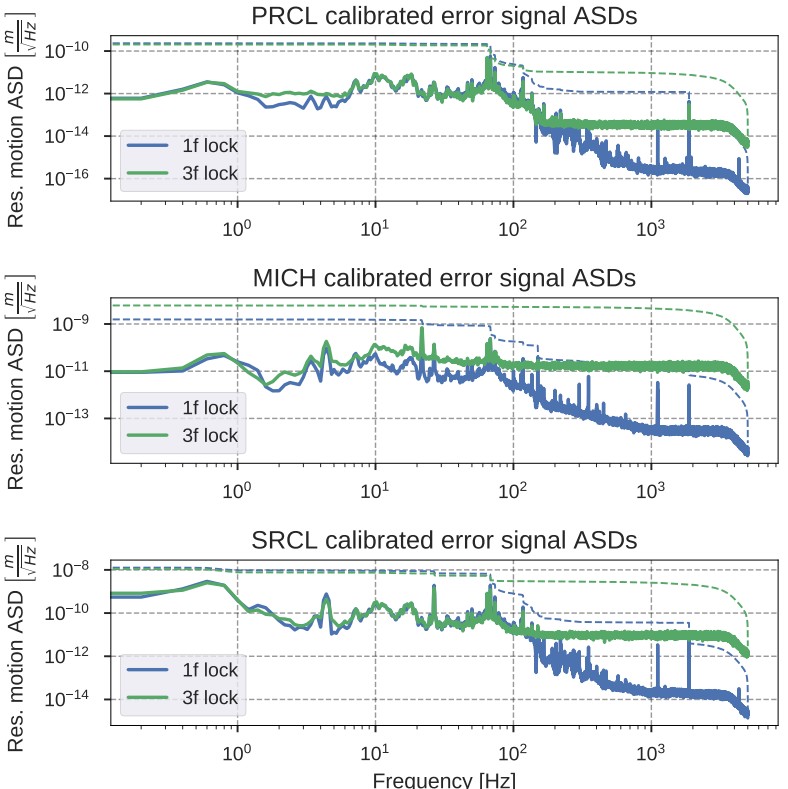

**Figure 3.** Experimental comparison between "1f" and "3f" error signals used to lock the DRMI. In particular, the comparison is between B2 6 MHz and B2 18 MHz for PRCL, and between B2 56 MHz and B2 169 MHz for SRCL and MICH. One can easily note that in all three cases, a flat shot noise dominates the spectrum above $\simeq$100 Hz in the "3f" signals. The plotted data has been acquired during the pre-O4 commissioning period, with 33 W of input power.

*Linearization of the PRCL Error Signal*

In a standard Fabry-Pérot cavity setup, power signals can be used to expand the linear region of a PDH-like error signal [12]. This is important during the lock acquisition because it increases the range inside which the controller can be engaged (more details in Section 3). In the case of the PRCL error signal, denoted by $e_{\text{PRCL}}$, a linearization strategy has been studied via numerical modeling with FINESSE, as shown in Figure 4a. The designated linearizing signal was the one that maximized the width of the linear region: the beat between the upper and lower 6 MHz sidebands at the pick-off in the PRC, namely B4 12 MHz. The linearized error signal is given as:

$$E_{\text{PRCL}} = \frac{\text{B2 6MHz}}{\text{B4 12MHz}}. \tag{4}$$

This strategy has been tested and implemented experimentally, as data are showing in Figure 4b.

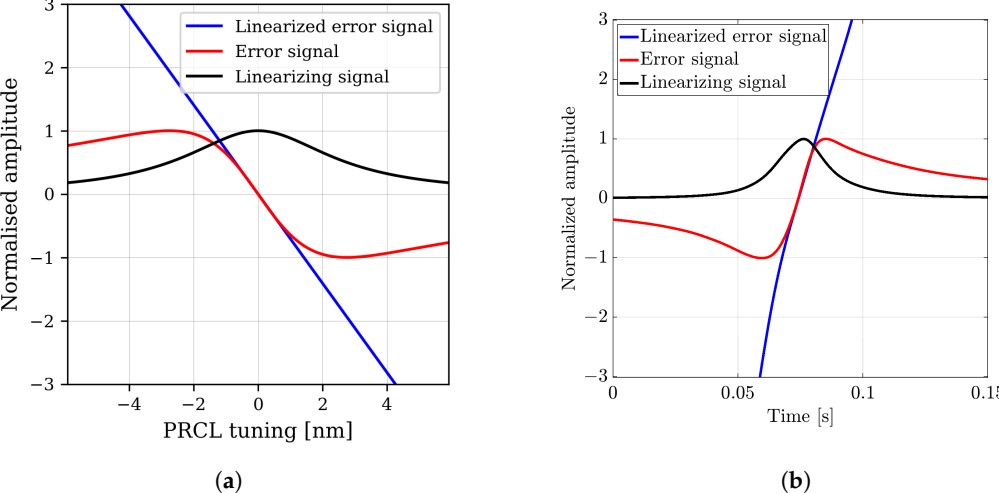

(**a**)                                                                                                       (**b**)

**Figure 4.** (**a**): Simulated error signal linearization. In red, the PRCL error signal, B2 6 MHz. In black, the linearizing signal, magnitude of B4 12 MHz. In blue, the PRCL error signal divided by the linearizing signal. (**b**): Experimental comparison, during a free scan of the PRCL DOF, of linearized and non-linearized signals for the control of PRCL in the proximity of the PRC resonance.

### 3. Trigger Logic for the Lock Acquisition

As introduced in Section 1, for the O4 Run, the lock acquisition of the entire interferometer has been changed and follows a new scheme. In particular, green lasers are used to keep the arm cavities outside the resonance of the main beam to ease the control of the DRMI [11]. The challenging part is that the two recycling cavities are intertwined as, e.g., the 56 MHz sideband resonates in both of them; therefore, the lock of the three DOFs related to the DRMI configuration (MICH, PRCL, and SRCL) has to happen *simultaneously* ("coincidence lock", cfr. Section 1).

For this technique to be effective, one must rely on a good trigger signal that guarantees that the interferometer is passing through a good working point, where the lock can be achieved in a statistically successful way. Attempting the lock in a random working point would be not only unsuccessful but also detrimental, as the force exerted on the optics' suspensions for achieving control would excite them, making following attempts more difficult. Therefore, it is of paramount importance to find a viable trigger for the lock acquisition attempt. Given that the DRMI involves three DOFs and five optics, a single signal cannot be used; instead, a combination must be found based on the available probes in the central area of the interferometer, each one sensitive to different DOFs.

This has been studied using a time domain model of the DRMI (without the End mirrors), with the E2E [10] simulation software. The model includes: a simplified injection system (main laser and sidebands generation); the mechanics of the suspension system for the involved mirrors, described as transfer functions; the optical parameters of the mirrors; lastly, all the relevant sensors (at DC and demodulated at the sidebands frequencies) of the central part of the interferometer. The model has not been used to simulate the lock acquisition itself but to investigate the behavior of the trigger candidates while the interferometer moves freely around the locked position.

The methodology for searching the trigger is, therefore, the following:

1. The simulated DOFs are put in the foreseen working point in order to assess the behavior of the signals from the probes in the working point and its vicinity. This defines the expected values of the several signals (Figure 5);

2. The aim is then to use only the information coming from the different probes to reconstruct as accurately as possible the working point. Given the relatively high number of DOFs, one single rule will not be enough for the determination of the

working point; instead, adding one rule after the other, an inverse selection rules system is designed, constraining the working point of the several DOFs more and more as additional signals are used;

3.   The goodness of the trigger will depend on whether it ensures the correct selection of the wanted working point without leading to false positives.

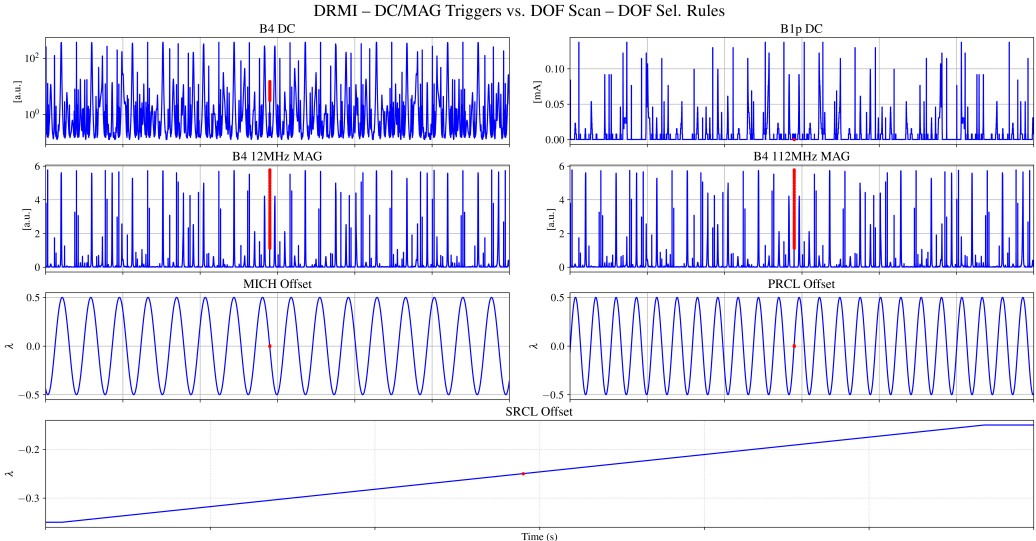

**Figure 5.** DRMI triggers: determination of the signals with respect to the DOF working point. This figure (shared *x* axis among subplots) represents a scan of the three DRMI DOFs, where it is known where the good working point is; in this way, it is possible to determine the value that signals of interest have in such point. During the study, a larger set of signals to be tested has been used to determine the best triggers; in this plot, only the most relevant signals are shown.

A through study on the best possible candidate signals was performed, aiming at the smallest possible complete set of inverse selection rules which can guarantee a correct estimation of the DOFs working point. The number of signals has been progressively reduced to four, and from each one, a selection rule has been computed, using adequate thresholds (either greater than or lower than) obtained from experimental data and based on normalized signals (so, in the $[0, 1]$ range); finally, their product forms the single final trigger, used for the lock acquisition of all the three DOFs of the DRMI configuration (Figure 6). The summary of these selection rules is the following:

- B4_12MHz_mag: beating of the 6 MHz upper and lower sidebands, probed in the Power-Recycling cavity; high values are expected when the three DOFs are on the working point since it is a measurement of the 6 MHz power (a value $\geq 0.7$ has been chosen);

- B1p_DC: DC of the dark port pick-off after the Signal-Recycling cavity; low values are expected when the three DOFs are on the working point (a value $\leq 0.1$ has been chosen);

- B4_112MHz_mag: beating of the 56 MHz upper and lower sidebands, probed in the Power-Recycling cavity; high values are expected when the three DOFs are on the working point since it is a measurement of the 56 MHz power (a value $\geq 0.6$ has been chosen);

- B4_DC: DC of the pick-off probed in the Power-Recycling cavity; low values are expected when the three DOFs are on the working point, as we want to lock on the sidebands and not on the Carrier (a value $\leq 0.1$ has been chosen).

This trigger choice has also been studied using FINESSE [8,13] to check that when the trigger thresholds are reached, all the error signals are in (or close to) their linear range. Figure 7 shows the control error signals while scanning the longitudinal DOFs. The black line represents the longitudinal tunings for which the trigger logic is met and control servos

can be engaged. The bottom row is a cross-section of the plots above to help visualize that error signals are sufficiently linear within this region. The results found are consistent with the time domain study.

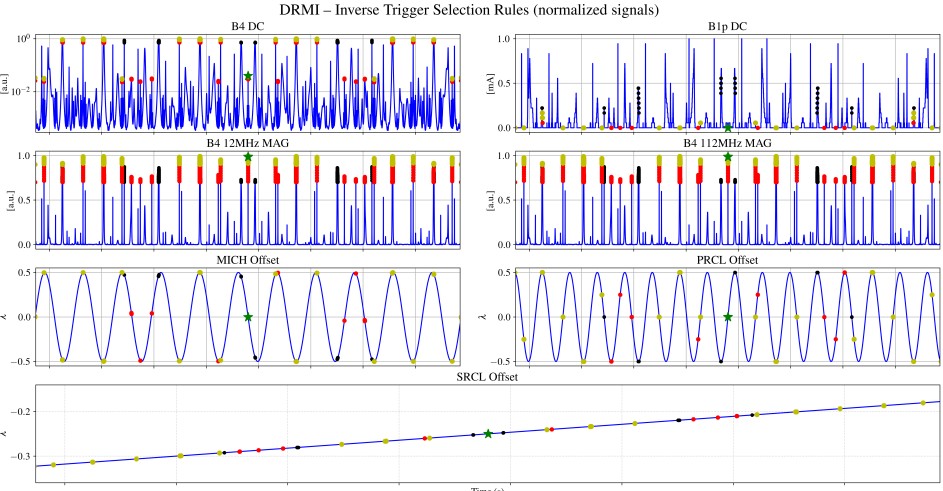

**Figure 6.** DRMI Triggers: this figure (shared *x* axis among subplots) represents another, different scan of the three DRMI DOFs, where it is known where the good working point is; such knowledge is not used, while instead inverse selection rules, based on the values of the signals of interest, are used to reconstruct the wanted working point; different, more stringent inverse selection rules are stacked from black to red, yellow and finally green until the correct working point is finally reconstructed.

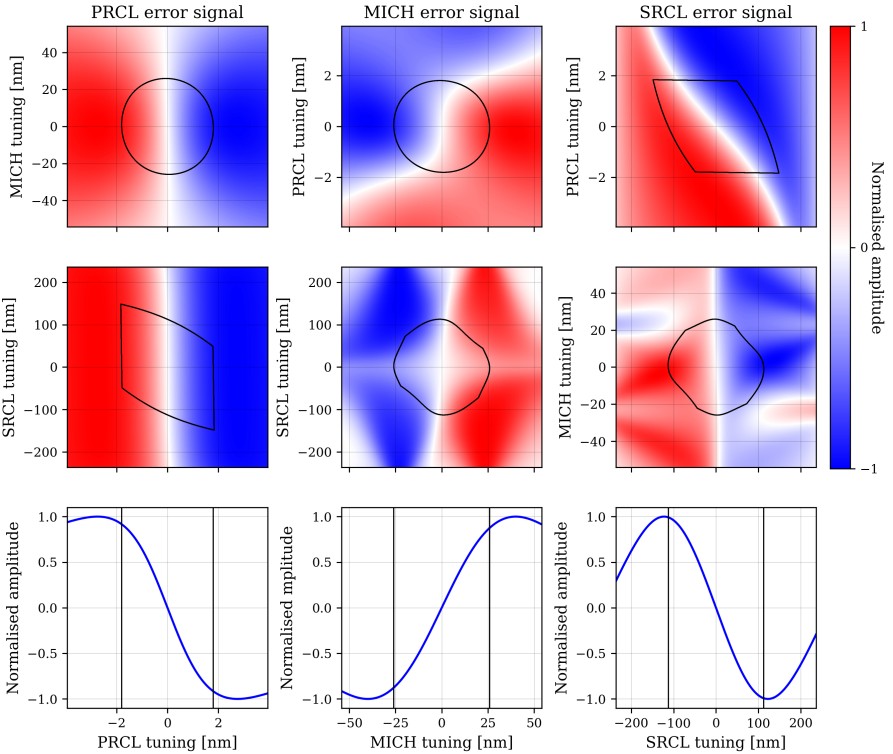

**Figure 7.** In the first two rows, error signals are plotted against the tuning of pairs of DOFs. At the center of each image is the operating point. Triggering thresholds are exceeded for the points within the black line. In the bottom row, a cross-section of the graphs above.

In Figure 8, we can see an example of a successful lock acquisition of the DRMI obtained by using the trigger computed with this technique in the real detector.

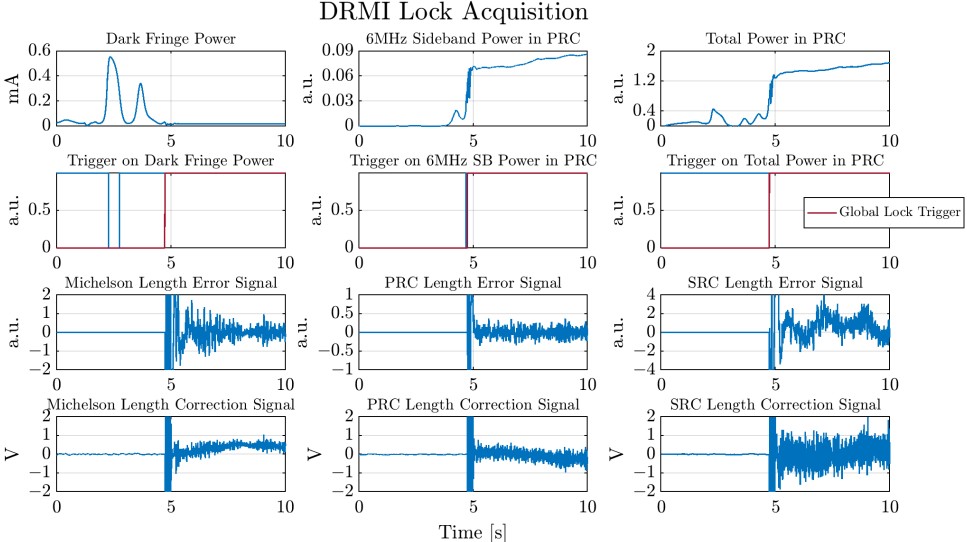

**Figure 8.** Example of DRMI lock acquisition during the commissioning of AdV+. In the second row there are, in blue, the three single trigger signals, based on the corresponding signals of the first row; in red, the final global trigger, computed as the product of the single triggers.

## 4. Effects of Misalignments on the Longitudinal Error Signals

During the acquisition of the DRMI lock, the mirror alignment is controlled by limited-bandwidth local controls, resulting in a residual angular motion of the order of magnitude of 0.05 µrad. The main problem is that the local controls error signals do not have information about the beam/mirror misalignment, but they are only a ground-based reference. This means that, even if the accuracy control is enough to maintain the lock, a pre-alignment strategy could be needed. Moreover, this amount of residual motion is incompatible with the accuracy required for the detection of gravitational waves, and, for this purpose, higher performance angular controls need to be enabled at the end of the lock acquisition [14]. A study of the error signals in the previous Power-Recycled Michelson Interferometer (PRMI) configuration evaluated the impact of misalignments of the PR mirror during the lock acquisition [15], crucial for the Variable Finesse technique [5].

In the new DRMI configuration, however, the lock of the DRMI DOFs is radically different from before. The structure of the new lock acquisition procedure raised concerns regarding the feasibility of achieving and then maintaining the DRMI lock for the full duration of the CARM offset reduction procedure, which lasts several minutes. During this period, a recycling cavity's alignment drift affecting the longitudinal error signals could ultimately spoil the stability of the lock.

Therefore, we simulated the influence of misalignments of the arm input mirrors, PR, and SR mirrors on the DRMI longitudinal error signals. We omit the results for the misalignment of the WI mirror, having an analogous behavior to the NI mirror due to the symmetry in the interferometer setup.

Particular focus was given to finding a maximum allowable misalignment for each mirror, above which the stability of the lock cannot be guaranteed. The full results of the simulations are reported in [16].

Simulating both the "1f" and "3f" error signals allowed us to respectively evaluate the possibility of engaging the lock and its robustness during the CARM offset reduction. The maximum allowable misalignment thresholds were then used to choose the alignment strategy during the CARM offset reduction phase.

### 4.1. Methodology

We simulated the misalignment effects using the modal simulation software FI-NESSE [8]. To optimize the simulation process, the procedure was divided into the following steps:

- Base simulation of misaligned interferometer and convergence check;
- Simulation of the longitudinal DOF sweeps;
- Optimization and study of the error signals.

These steps will be explained in detail in the next Section.

### 4.2. Simulation of Longitudinal Detunings

To accurately study the behavior of the interferometer when subject to misalignments, the correct working points of the longitudinal DOFs need to be found for each angle of each misaligned mirror. This is accomplished by minimizing the "3f" error signals using the built-in "lock" command of FINESSE. The minimization simulates the action of the DRMI longitudinal loops, which are indeed engaged during the simulated scenario.

During this step, the convergence of the simulations is also tested. This means estimating whether the amount of Transverse Electro-Magnetic (TEM) Higher-Order Modes (HOMs) used in the simulation is sufficient to give realistic results. The convergence has been checked by finding the DRMI working points with an increasingly higher number of HOMs and then comparing the results.

The results show that using as maximum HOM order 14; the PR misalignment simulations converge up to $\simeq 1.5 \cdot 10^{-7}$ rad. In the same way, the convergence of the NI misalignment was obtained up to $\simeq 1.8 \cdot 10^{-7}$ rad. For the misalignment of SR, the simulations converged in all the simulated ranges, up to $\simeq 3 \cdot 10^{-6}$ rad, which are compliant with the local control accuracy.

The initial simulations allow us to study the error signals at their working point and to estimate their optical gain, optimal demodulation phase, and cross-couplings. However, further simulations are required to check how the misalignment of the mirror affects the error signals in a region around the working points. Therefore, we simulated longitudinal sweeps of one DRMI Longitudinal Sensing and Control (LSC) DOF at a time for a limited number of misalignments while keeping the others at the working point. Simulating these sweeps allows checking the shape and linearity of the error signals around the working point, including checking for multiple zero-crossings.

Additionally, the zero-crossings of the error signals can be compared with the resonance peaks of the Carrier and (more importantly for this step of the lock acquisition) of the control sideband fields. The figures of merit to be analyzed are:

- Optical gains;
- Working point;
- Shape of the error signal (multiple zero crossings, etc.).

### 4.2.1. Optical Gain and Optimal Demodulation Phase Variations

One of the main effects of the misalignment is a progressive reduction of the optical gains of the longitudinal PDH error signals. The extent of this effect depends both on the error signal and on which mirror is misaligned. As expected, the simulations show that PDH signals modulated at a higher frequency are overall less impacted by the misalignment than the lower frequency ones. This is due to the Schnupp asymmetry [17], which is a macroscopic arm length difference that allows the sideband fields to be partially transmitted towards the antisymmetric port while keeping the carrier at dark fringe condition. This asymmetry causes larger recycling losses in the higher-frequency sidebands, resulting in a higher Finesse for the lower-frequency ones. Finally, the losses given by cavity misalignments have a lower impact the more other loss sources (such as the Schnupp asymmetry) are relevant.

The amount of gain loss that can affect a loop before it loses lock depends on several factors, particularly on the "gain margin" of the involved control loop and the SNR of the error signal used.

It has to be considered that the standard gain margin of a longitudinal control often leads to a lock loss if an optical gain variation is above a factor of 2. For this reason, the misalignment angle at which this threshold is reached is an important figure of merit subject of this study. This corresponds to the maximum misalignment variation that can be tolerated to acquire the lock.

Figure 9 shows as an example the relative effect of PR mirror misalignments on the planned error signals' optical gains. The optical gains are normalized by the corresponding values in the aligned condition. The results of the complete study are visible in Table 2.

From the results, one can see that the most restrictive condition is given by the behavior of the B2 6 and 18 MHz signals (used to control PRC) for PR mirror misalignments. These error signals lose 50% of their optical gain at $5.7 \cdot 10^{-8}$ rad of PR misalignment (see Figure 9).

This error signal is also the most affected by NI mirror misalignments, with a half-gain threshold of $7.8 \cdot 10^{-8}$ rad. In both cases, the misalignment affects the 6 MHz and 18 MHz signals equivalently.

SR misalignments instead have a much lower overall impact on the error signals and, as expected, they have more effect on the 56 MHz sidebands signals than on the 6 MHz ones due to the 56 MHz sidebands being resonant in the SRC cavity. Additionally, the MICH error signal gain increases with the SR mirror misalignment. No maximum threshold has been found for the simulated misalignments. Increasing the simulation range will eventually find a threshold given by the gain drop of SRCL itself. However, this is not of interest for this study since the simulated misalignment is already well above the residual angular motion with the mirror under the local controls alone (0.05 µrad).

Another parameter that can make the lock fail is the deviation from the optimal demodulation phase, which will introduce cross-couplings. From the results of the simulations, the change in the optimal demodulation phase is always lower than 10° for misalignments within the thresholds shown in Table 2. Therefore, the change in the demodulation phase is not the limiting factor in the maximum allowed misalignment.

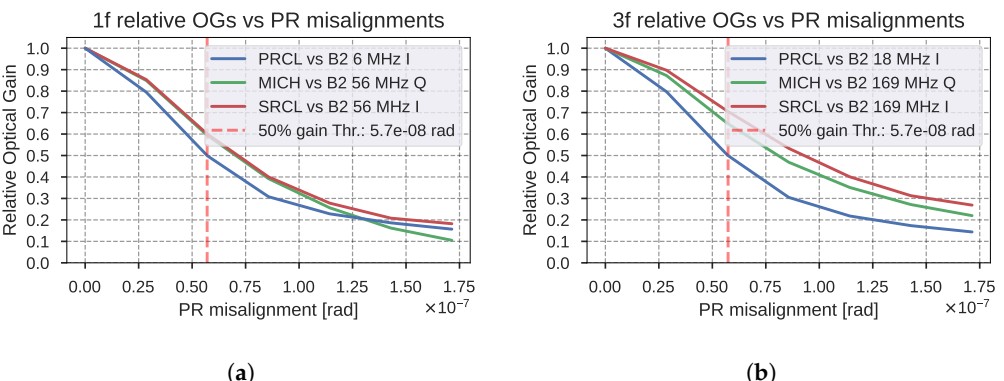

(**a**)　　　　　　　　　　　　　　(**b**)

**Figure 9.** DRMI error signals Optical Gain (OG) variation vs. misalignments of the PR mirror. Each line corresponds to the relative optical gain of an error signal used to control the DRMI. (**a**) shows the "1f" error signals; (**b**) shows the "3f" error signals. The OGs are relative to the gains in a fully aligned condition. The vertical line marks the lowest angle at which one of the error signals has lost 50% of its OG.

**Table 2.** 50% OG loss thresholds for each error signal. Each column shows the respective mirror misalignment at which the error signal loses 50% of its optical gain. For SR misalignments (last column), most of the error signals do not reach the 50% threshold in the simulated range (up to $3.0 \cdot 10^{-6}$ rad).

| DOF | Signal | PR Thr. [Rad] | NI Thr. [Rad] | SR Thr. [Rad] |
|------|--------------|--------------------|--------------------|--------------------|
| PRCL | B2 6 MHz I | $5.7 \cdot 10^{-8}$ | $7.8 \cdot 10^{-8}$ | N.A. |
|      | B2 18 MHz I | $5.7 \cdot 10^{-8}$ | $7.8 \cdot 10^{-8}$ | N.A. |
| MICH | B2 56 MHz I | $6.9 \cdot 10^{-8}$ | $1.0 \cdot 10^{-7}$ | N.A. |
|      | B2 169 MHz I | $8.0 \cdot 10^{-8}$ | $1.2 \cdot 10^{-7}$ | N.A. |
| SRCL | B2 56 MHz Q | $7.0 \cdot 10^{-8}$ | $9.6 \cdot 10^{-8}$ | N.A. |
|      | B2 169 MHz Q | $9.2 \cdot 10^{-8}$ | $1.2 \cdot 10^{-7}$ | $2.4 \cdot 10^{-6}$ |

### 4.2.2. Cross Couplings

Another effect that has been studied is the influence of misalignments on the couplings between different DOFs. For the misalignment of each mirror, we compared the optical gain of each error signal against displacements of the three different DRMI longitudinal DOFs.

The results show that PRCL has a large impact on all the error signals and that its error signals (B2 6 and 18 MHz) are minimally influenced by MICH and SRCL displacements, even at high misalignments. Nevertheless, due to its high optical gain (see Section 2), the PRCL loop can be closed with much higher bandwidth than the MICH and SRCL ones. This allows suppressing PRCL residual motion at the level of having little impact on MICH and SRCL. Indeed, experimental results confirmed that the PRCL loop is stable up to an Unity Gain Frequency (UGF) of $\simeq$70 Hz, while SRCL and MICH are respectively kept at UGFs of $\simeq$3 Hz and $\simeq$10 Hz during the lock acquisition. Additionally, the phase of the MICH error signal (B2 169 MHz I) is maintained as orthogonal as possible to the optimal PRCL phase, minimizing the influence of its displacement. This has the downside of maximizing the impact of PRCL displacements on the SRCL loop, but since this loop has the lowest bandwidth and the largest linear region, its robustness is not affected by this cross-coupling.

SRCL detuning has a low impact on all error signals for any misalignment angle, not raising particular concerns.

The MICH displacement impact on the PRCL error signal remains negligible. Its impact on the SRCL error signal (B2 169 Hz Q) instead increases by a factor of $\simeq$2 for PR and NI misalignments above $0.5 \cdot 10^{-7}$ rad. Considering that the MICH impact remains a factor 4 below the SRCL gain and considering the high robustness and low UGF of the SRCL loop, the main limiting factor to the NI and PR misalignments remains the loss of optical gain.

### 4.2.3. Error Signal Shapes and Zero-Crossing

The simulated DOF sweeps allowed us to check for abnormalities in the error signals' shape and the correspondence of their zero-crossings with the control sideband peaks. The results for the "3f" error signals used to control PRCL are represented in Figure 10.

No particular concerns arise from the shapes of the error signals. At the simulated angles, the shape and the linear region of all the error signals are minimally affected, with the loss of optical gain as the main noticeable effect. The most noteworthy zero-crossing deviation is the SRCL error signal (B2 169 MHz Q), which, at the maximum simulated SR mirror misalignment, deviates by $\simeq$1.7 nm from the 56 MHz sideband peak observed by the B2 112 MHz magnitude signal. However, as mentioned above, the maximum simulated SR misalignment ($\simeq$3 µrad) represents an unlikely scenario, even with the alignment controlled by local controls. Lower SR misalignments or large PR and NI misalignments also introduce a similar deviation but of negligible magnitude ($\leq$0.5 nm).

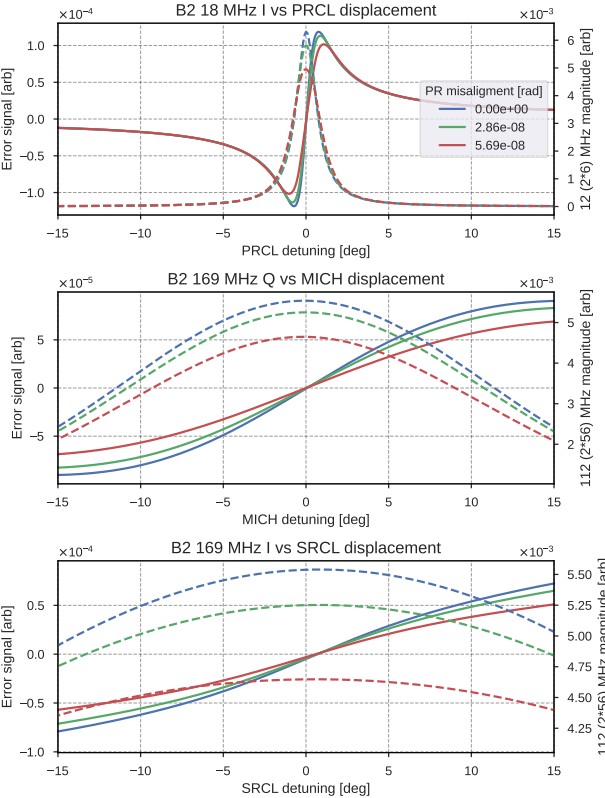

**Figure 10.** DRMI error signals scans for various PR misalignments. Each plot shows the scan of a different DRMI DOF and its corresponding "3f" error signal (continuous lines). The dashed lines show the "2f" signal magnitude , acquired on B4, corresponding to the resonances of either the 6 MHz (first plot) or 56 MHz (second and third plot).

### 4.2.4. Results

After having evaluated the effect of the misalignment of PR, NI, and SR mirrors, it can be concluded that the most critical locking parameter affected is the Optical Gain variation, as shown in Table 2.

From the results, it is clear that for both NI and PR mirrors, a pre-alignment strategy is required in order to be able to acquire the lock.

For the input mirrors, this pre-alignment is already obtained with the cavity pre-alignment procedure of the long arms, foreseen at each lock acquisition, based on the *dithering technique* [14].

Instead, for the PR mirror, we implemented a global pre-alignment algorithm aimed at maximizing the sidebands power inside the DRMI; however, since the chosen pre-alignment has very low accuracy, it is important to switch, after the lock is acquired, to a global signal for the control of the PR alignment.

The study aimed to determine the global PR alignment error signals will then be described in the next Section.

## 5. Alignment Controls

The outcomes of the study shown in Section 4 show the need for a global reference control for the PR mirror to ensure the stability of the longitudinal control. For this reason, an investigation of the possible global error signals for PR has been carried on, and the results will be described in this Section.

In AdV+, the deployed technique to build the error signals is based on the Ward technique [7], which requires two Quadrant PhotoDiodes (QPDs), with their Gouy phases 90° apart, and demodulated at frequencies that anti-resonate in the cavity of interest. Since the laser field already contains modulation frequencies used for longitudinal controls, as

mentioned in the previous Sections, the additional hardware needed to implement the Ward technique is limited only to the QPDs and the telescopes to set the proper Gouy phases.

For the DRMI of AdV+, the available ports to attach the QPDs to are illustrated in Figure 11.

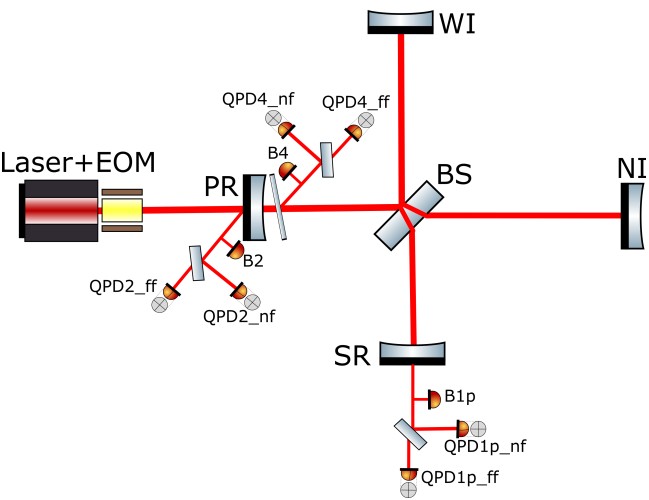

**Figure 11.** Simplified scheme of the optical layout of the DRMI with the QPDs used for the automatic alignment sensing.

Each of the quadrants shown in Figure 11 generates both DC signals, which are simply given by the difference of the power detected on each half of the quadrants, and demodulated signals (obtained with the Ward technique), which provide the wavefront sensing. The demodulation happens at the same frequencies used for the longitudinal control.

Given the complexity of the optical layout of the DRMI, the possible candidate error signals for the automatic alignment are investigated through numerical simulations rather than analytical computation. The chosen tool for this is FINESSE [8]. Since misaligning a mirror introduces a variation in the length of the corresponding cavity, the cavity itself needs to be controlled in order not to change its resonance conditions. This can be achieved by correcting, for each step of the angular scan, the longitudinal positions of the DRMI mirrors using the information from the longitudinal error signals. See Appendix A for a list of the most relevant parameters of the simulated interferometer.

*Simulation Results*

An in-depth analysis of these results can be found in [18]. In general, the interesting signals must have a steep slope around the zero-crossing (also referred to as optical gain), it must not present multiple zero-crossings, and it must not observe the alignment of multiple mirrors with comparable intensities.

For the DRMI obtained with the CARM offset, it has been found that the PR mirror generates a good signal on B2 Near-Field (NF) 6 MHz, as shown in Figure 12. It also shows, in dashed style, the signals for NI and WI mirrors to indicate that, in this configuration, they are to be considered aligned, with their contribution to the total signal on the sensor to be neglected. Without this assumption, their signal would be strongly coupled with each other's and with the PR mirror's one since the Ward signals have very similar amplitude and optical gain, and the optimal demodulation phase is the same for the three of them. This assumption can be indeed considered true since the long arm cavity mirrors are pre-aligned with the *dithering technique* before engaging the DRMI lock [14].

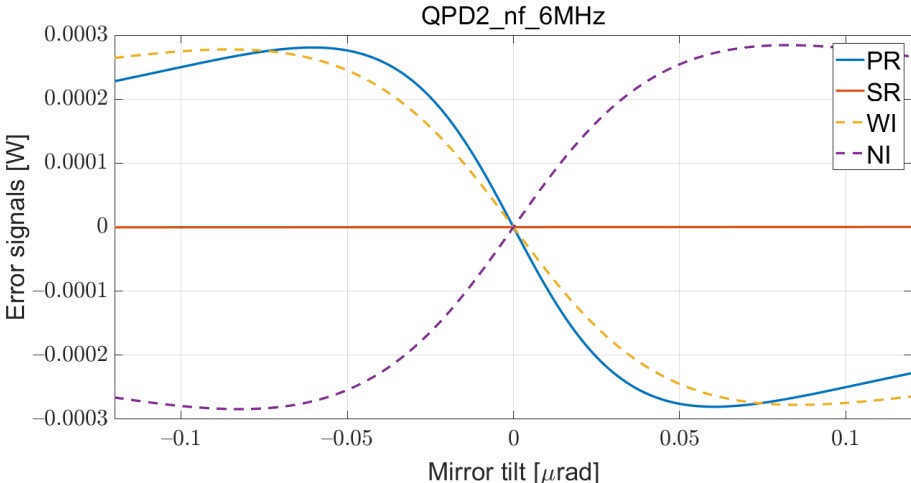

**Figure 12.** Signals (**left**) and compass plot (**right**) for B2 NF 6 MHz comparing the signals produced by the misalignment of PR (blue), SR (red), WI (yellow, dashed), and NI (purple, dashed) mirrors [18].

The following Table 3 shows the optical gain of the simulated signal, expressed in dB with respect to the smallest one, and the optimal demodulation phase:

**Table 3.** Optical gains (in dB, normalized over the smallest) and demodulation phases (in degree) of the simulated signals shown in Figure 12.

| DOF | Mag | Phase |
| --- | --- | --- |
| PR | 72.80 dB | 89.10° |
| SR | 0.00 dB | −78.30° |
| NI | 70.02 dB | 89.10° |
| WI | 70.05 dB | 89.10° |

In Table 3 we can especially notice how the optimal demodulation phase for the signal generated by PR, NI and WI (not counting a difference of ±180°) is the same, and the optical gains are very close to each other. This is an example of the widespread coupling of the different DOF mentioned above.

The optimal error signal for the alignment of the PR mirror has been tested and successfully implemented in the real detector.

Figure 13 shows the moment the PR alignment loop is engaged on this signal: the pitch and yaw components are shown in the top row, and the effects on two relevant figures of merit (sideband power and total intracavity power in the PRC) are shown on the bottom row.

In Figure 13, one can notice that, as the error signals get closer to zero, the sidebands' powers and the intracavity power reach their maximum.

Finally, since the SR generate signals that is always overwhelmed by the other DOFs on every QPD, it is not possible to isolate a good candidate to feed an alignment loop. For the lock of the DRMI, its alignment is given by optical levers impinging on PSD that are connected to the ground [19], so that their reference frame is given by the ground itself, rather than the beam circulating in the interferometer.

Finally, as it was done for the longitudinal control signals, after the lock acquisition of the DRMI, the angular control of the PR is handed-off to its "3f"-demodulated sensor.

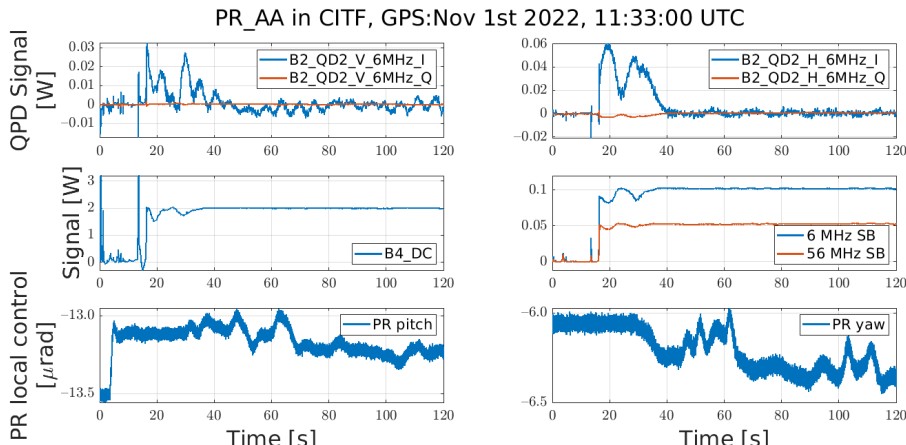

**Figure 13.** Example of the engagement of the angular control of the PR mirror after the lock of the DRMI. The top plots show the I and Q projections of the angular error signals for yaw (horizontal) and pitch (vertical) DOFs of the PR mirror. The Q is almost flat, signaling a good tuning of the demodulation phase. The middle plots show the behavior of the global power (left) and the sidebands' power (12 MHz and 112 MHz, right) picked-off in the PRC. The bottom plots show the pitch (left) and the yaw (right) of the PR mirror as detected by the optical levers for its marionetta.

## 6. Conclusions

The studies presented in this article allowed us to design and validate the lock strategy of the DRMI configuration. In particular, the results allowed us to determine the following crucial points:

- The error signals used to lock the longitudinal loops for both the initial acquisition and the intermediate state during the CARM offset reduction;
- The trigger signals used to determine the good conditions for the engagement of the feedback loops;
- The robustness of the chosen longitudinal error signals and the necessity of a global angular control of the PR mirror in order to maintain the lock during the whole lock acquisition procedure;
- The error signal for the aforementioned angular control of the PR mirror.

Subsequently, the new lock strategy was implemented on the upgraded Advanced Virgo Plus detector. The commissioning of the DRMI started in late February 2021, and its lock was achieved after less than one month. Despite the limitations given by the non-optimally matched recycling cavities, the results of the commissioning confirmed the validity of the simulation studies. Indeed, this part of the lock acquisition remains fundamentally unchanged after more than one year of tuning and optimization of the overall procedure.

The methodologies explained in this paper showed therefore, particular robustness and are currently being re-applied in the preparation of the "Phase II" of the Advanced Virgo Plus project, which will include further upgrades in the optical setup of the interferometer.

**Author Contributions:** Conceptualization, M.M.; methodology, D.B., M.B., J.C.D., A.F., R.M., M.M. and M.V.; software, D.B., M.B., J.C.D., A.F., R.M. and M.V.; validation, D.B., M.B., J.C.D. and M.M.; formal analysis, D.B., M.B., J.C.D., R.M. and M.V.; investigation, D.B., M.B., J.C.D., R.M., M.M. and M.V.; resources, A.F.; writing—original draft preparation, D.B., M.B., J.C.D., R.M. and M.V.; writing— review and editing, D.B., J.C.D., M.M. and M.V.; visualization, D.B., M.B., R.M., M.M. and M.V.; supervision, M.M. All authors have read and agreed to the published version of the manuscript.

**Funding:** This research received no external funding.

**Data Availability Statement:** Not available. This is mostly a simulation paper; the very little amount of real data used is not publicly available, as an open data policy is not enforced for the Commissioning phases of the detector. Real data, with the same validity regarding the assessment of the simulations here reported, will be available after the standard disclosure time period foreseen after the start of the O4 Run.

**Acknowledgments:** The authors gratefully acknowledge the Italian Istituto Nazionale di Fisica Nucleare (INFN), the French Centre National de la Recherche Scientifique (CNRS) and the Netherlands Organization for Scientific Research (NWO), for the construction and operation of the Virgo detector and the creation and support of the EGO consortium. The authors also gratefully acknowledge research support from these agencies as well as by the Spanish Agencia Estatal de Investigación, the Consellera d'Innovació, Universitats, Ciència i Societat Digital de la Generalitat Valenciana and the CERCA Programme Generalitat de Catalunya, Spain, the National Science Centre of Poland and the European Union—European Regional Development Fund; Foundation for Polish Science (FNP), the Hungarian Scientific Research Fund (OTKA), the French Lyon Institute of Origins (LIO), the Belgian Fonds de la Recherche Scientifique (FRS-FNRS), Actions de Recherche Concertées (ARC) and Fonds Wetenschappelijk Onderzoek–Vlaanderen (FWO), Belgium, the European Commission. The authors gratefully acknowledge the support of the NSF, STFC, INFN, CNRS and Nikhef for provision of computational resources. A particular acknowledgment goes to Priyanka Giri, Manuel Pinto and Enzo Tapia, who extensively contributed to the DRMI commissioning activities.

**Conflicts of Interest:** The authors declare no conflict of interest.

## Abbreviations

The following abbreviations are used in this manuscript:

| | |
|---|---|
| AdV+ | Advanced Virgo Plus |
| BNS | Binary Neutron Star |
| BS | Beam-Splitter |
| CARM | Common ARM |
| DOF | Degree Of Freedom |
| DRFPMI | Dual-Recycled Fabry-Pérot Michelson Interferometer |
| DRMI | Dual-Recycled Michelson Interferometer |
| EOM | Electro-Optic Modulator |
| HOM | Higher-Order Mode |
| INFN | Istituto Nazionale di Fisica Nucleare |
| IR | InfraRed |
| ITF | InTerFerometer |
| LIGO | Laser Interferometer Gravitational-Wave Observatory |
| LSC | Longitudinal Sensing and Control |
| MICH | short MICHelson length |
| NE | North-End |
| NF | Near-Field |
| NI | North Input |
| OG | Optical Gain |
| PDH | Pound-Drever-Hall |
| PR | Power-Recycling |
| PRC | Power-Recycling Cavity |
| PRCL | Power-Recycling Cavity Length |
| PRMI | Power-Recycled Michelson Interferometer |
| QPD | Quadrant PhotoDiode |
| SNR | Signal-to-Noise Ratio |
| SR | Signal Recycling |
| SRC | Signal-Recycling Cavity |
| SRCL | Signal-Recycling Cavity Length |
| TEM | Transverse Electro-Magnetic |
| UGF | Unity Gain Frequency |
| WE | West-End |
| WI | West Input |

## Appendix A. Parameters of the Simulations

These are the parameters set in the simulated interferometer. The radius of curvature ($RoC$), transmissivity $T$ and losses $L$ of the mirrors of the DRMI are as follows:

**Table A1.** Parameters of the mirrors of the simulated DRMI.

| Mirror | $RoC$ [m] | $T$ | $L$ |
|--------|-----------|-----|-----|
| PR | 1430 | 0.048 35 | $3.0 \cdot 10^{-5}$ |
| SR | 1430 | 0.4 | $3.0 \cdot 10^{-5}$ |
| WI | 1424.6 | 0.013 75 | $2.7 \cdot 10^{-5}$ |
| NI | 1424.5 | 0.013 77 | $2.7 \cdot 10^{-5}$ |

Other interesting parameters are the power $P_0$ injected in the simulated interferometer and the length of the recycling cavities:

- $P_0 = 40$ W
- $l_{PRC} = l_{PR} + \frac{l_{NI} + l_{WI}}{2} = 11.245$ m
- $l_{SRC} = l_{SR} + \frac{l_{NI} + l_{WI}}{2} = 11.248$ m

The modulation frequencies and depths are set as follows:

**Table A2.** Modulation frequencies and depths for the simulated control.

| $f_{mod}$ [Hz] | Depth |
|----------------|-------|
| 6,270,777 | 0.22 |
| 8,361,036 | 0.15 |
| 56,436,993 | 0.16 |

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
