# Peer review of "Simulations for the Locking and Alignment Strategy of the DRMI Configuration of the Advanced Virgo Plus Detector"

_galaxies, doi:10.3390/galaxies10060115_

Round 1
Reviewer 1 Report
This is an excellent, comprehensive study of the optical simulations used in the design and the commissioning of Dual-Recycled Michelson Interferometer of AdVirgo+ configuration.
Author Response
Thanks, much appreciated.
Reviewer 2 Report
This paper summarises the control of the DRMI degrees of freedom in the Advanced Virgo Detector. The authors use simulations to optimize the DRMI locking procedure by selecting appropriate sensors for various control loops, investigating an optimal lock trigger, studying the effects of misalignment on the error signals, exploring various cross couplings between the different longitudinal degrees of freedom, and exploring the angular control of the DRMI cavities. The paper is concise and explains DRMI’s controls well and is fit for publication after a few modifications. The following comments are prefaced by the line numbers in the text.
-
51: It is a bit unclear whether the previous ‘Variable Finesse’ acquisition refers to lock acquisition for all degrees of freedom in the interferometer or just PRMI (given that previous configurations did not have an SRC)
-
85: It is not entirely obvious that the carrier being resonant in the PRC follows CARM offset reduction. Is the CARM offset reduced by moving the frequency of the carrier? Or is it done by moving WI and NI (and consequently PRCL) in order to be resonant for the carrier?
-
109: Consider rephrasing to ‘that does not saturate the actuators’. The current phrasing makes it sound a bit unclear.
-
115: It would be useful to cross-reference Figure 1 while mentioning B2. Also it would be good to provide some intuitive explanation as to why the B2 sensor has the highest SNR and there should also be a mention of which other sensors were considered in the analysis.
-
123: Please specify that the 3f signal of interest is the beatnote between the 1f and 2f harmonics from the EOM rather than the 3f and carrier.
-
135: Similar to comment #4, the phrasing suggests that other linearizing signals could have been considered. In that case, it would be good to mention which other signals were considered and intuitively explain why POP is the best.
-
For Figure 3, what input power do these plots correspond to? Also, is this simulated data or experimental data?
-
178 : Through -> thorough
-
190 : B1 doesn’t seem to be shown in Figure 1.
-
Figure 5. Unclear that the red points/lines correspond to the same times or whether the plots share the same axes. Also, Figs 5 and 6 are very similar. Consider condensing into a single Figure
-
284 : This line implies that the optical gain of a higher finesse cavity is less sensitive to misalignment. Maybe one line explaining why that’s true would be useful
-
Section 4.2.2 talks about a bunch of cross couplings between different DRMI DoFs. While this is quite interesting and parts of it are understandable intuitively, it would be helpful to include a few plots such as transfer functions between MICH/SRCL/PRCL based on a model of the loops that are being used.
-
Figure 10. Just to be clear, the 2f signals are being measured on B4?
-
Figure 12. There are two sets of purple/yellow dashed traces. One set is consistent with the text mentioning that NI and WI couple strongly to the alignment signal. The other set however, are very low gain signals so it’s pretty unclear what they’re supposed to be.
-
435: Please expand PSD, cite optical lever control.
-
A table with all the different parameters of the simulations would be very useful. This could include the input power, PRM and SRM transmissivities, PRC and SRC length, modulation depths of the RF sidebands.
Author Response
We agreed with the vast majority of the reviewer observations. Please see attachment for point by point answers.
